# A Data-Driven Approach to Construct a Molecular Map of *Trypanosoma cruzi* to Identify Drugs and Vaccine Targets

**DOI:** 10.3390/vaccines11020267

**Published:** 2023-01-26

**Authors:** Swarsat Kaushik Nath, Preeti Pankajakshan, Trapti Sharma, Priya Kumari, Sweety Shinde, Nikita Garg, Kartavya Mathur, Nevidita Arambam, Divyank Harjani, Manpriya Raj, Garwit Kwatra, Sayantan Venkatesh, Alakto Choudhoury, Saima Bano, Prashansa Tayal, Mahek Sharan, Ruchika Arora, Ulrich Strych, Peter J. Hotez, Maria Elena Bottazzi, Kamal Rawal

**Affiliations:** 1Centre for Computational Biology and Bioinformatics, Amity Institute of Biotechnology, Amity University, Noida 201303, Uttar Pradesh, India; 2Texas Children’s Hospital Center for Vaccine Development, Departments of Pediatrics and Molecular Virology and Microbiology, Baylor College of Medicine, Houston, TX 77030, USA; 3National School of Tropical Medicine, Baylor College of Medicine, Houston, TX 77030, USA; 4Department of Biology, Baylor University, Waco, TX 76798, USA

**Keywords:** system biology, vaccine targets, pathways, gene ontology, chagas disease, *Trypanosoma cruzi*, drug target

## Abstract

Chagas disease (CD) is endemic in large parts of Central and South America, as well as in Texas and the southern regions of the United States. Successful parasites, such as the causative agent of CD, *Trypanosoma cruzi* have adapted to specific hosts during their phylogenesis. In this work, we have assembled an interactive network of the complex relations that occur between molecules within *T. cruzi*. An expert curation strategy was combined with a text-mining approach to screen 10,234 full-length research articles and over 200,000 abstracts relevant to *T. cruzi*. We obtained a scale-free network consisting of 1055 nodes and 874 edges, and composed of 838 proteins, 43 genes, 20 complexes, 9 RNAs, 36 simple molecules, 81 phenotypes, and 37 known pharmaceuticals. Further, we deployed an automated docking pipeline to conduct large-scale docking studies involving several thousand drugs and potential targets to identify network-based binding propensities. These experiments have revealed that the existing FDA-approved drugs benznidazole (Bz) and nifurtimox (Nf) show comparatively high binding energies to the *T. cruzi* network proteins (e.g., PIF1 helicase-like protein, trans-sialidase), when compared with control datasets consisting of proteins from other pathogens. We envisage this work to be of value to those interested in finding new vaccines for CD, as well as drugs against the *T. cruzi* parasite.

## 1. Background

Chagas disease is caused by the protist parasite *Trypanosoma cruzi*. It affects 6–7 million humans and a large number of animal species. The study of CD and *T. cruzi* is challenging, due to the complexity and unique characteristics of the parasite’s genome. For instance, 50% of the *T. cruzi* genome is composed of repeated sequences, such as transposable elements, microsatellites, and simple tandem repeats. It also includes surface molecules encoding genes, such as trans-sialidases, mucins, gp63s, and a large novel family (>1300 copies) of the mucin-associated surface protein (MASP) [1,2,3].

*T. cruzi* presents with a complex life cycle comprising four morphological stages: epimastigotes (EP), metacyclic trypomastigotes (MT), cell-derived trypomastigotes (CDT), and amastigotes (AM). During its life cycle, the parasite changes in morphology, metabolism, and gene expression, as it passes from the epimastigote replicative stage in the insect to the metacyclic trypomastigote form, which infects humans. *T. cruzi* appears to have a unique approach to evading both parasiticidal antibodies and T-cell recognition. It creates an antigenic variation by generating large families of genes encoding both surface and secreted proteins, many of which are expressed simultaneously. Previously, Nde et al., first elucidated the lamc1 sub-network interactome mobilised in the human coronary artery smooth muscle (HCASM) cells by *T. cruzi*. The authors presented the human extracellular matrix interactome network regulated by *T. cruzi* and its gp83 ligand that facilitates cellular infection [4]. These authors also reported that *T. cruzi* modulates the extracellular matrix (ECM) interactome to gain cell entry while evading the host immune system [4]. *T. cruzi* triggers gp83 receptors in the host cell via ERK1/2 to up-regulate LAMC1 which cross-talks to both *LGALS3* and *THBS1* to enhance cellular infection using selected parasite surface molecules, such as TcCRT and TC45 mucin [4]. Roberts et al. used proteomics and interaction data to generate constraint-based models, focusing on pathways of the central metabolism, such as glycolysis [5]. Researchers have also constructed genome-scale metabolic models of *T. cruzi* to investigate the variation in metabolic functions across the life cycle stages and to predict the stage-specific essential genes and reactions. These reactions are essential for the growth of *T. cruzi* and could potentially be drug targets, such as the cytosolic transketolase reaction, the mitochondrial alanine transport reaction, and the glycosomal glucokinase reaction [6].

Despite these efforts, there remains an urgent need to construct a molecular network of *T. cruzi* to understand the pathophysiology of trypanosome infection. Networks have played an important role in the understanding of complex diseases, such as obesity, atherogenesis, myocardial infarction, heart failure, Charcot–Marie–Tooth disease, and spinal muscular atrophy [7,8,9,10,11,12,13]. It has been proposed that pathogens that cause acute infections tend to target central hubs of the host’s cellular signaling networks, whereas pathogens causing chronic disease are more likely to target peripheral nodes [14]. Furthermore, the network pharmacology approach has been used to study “compound-protein/genes-disease” pathways, which can describe complexities among biological systems, drugs, and diseases. Thus, molecular networks provide significant avenues for the identification of drug targets [15,16,17], vaccine targets [18,19,20], and biomarkers [21,22,23].

Here, we present a molecular map of *T. cruzi* constructed from an extensive literature review using deep curation and text-mining techniques. We also conducted large-scale docking studies of benznidazole and nifurtimox against several thousand *T. cruzi* predicted protein structures using our new drug-dock approach to identify the important molecular targets of Bz and Nf in the *T. cruzi* molecular network. This work also identified other FDA-approved drug molecules, in the context of *T. cruzi*, and new strategies to identify drug targets using comparative approaches. 

## 2. Methods

A hybrid approach combining deep curation and text-mining strategies was used to curate the relevant information related to the *T. cruzi* pathways and molecules (genes, proteins, drugs, etc.) from published scientific articles. The abstracts were retrieved from literature databases, such as PubMed and Google Scholar using keywords, such as “*T. cruzi*” AND pathways name, “*T. cruzi* AND molecules (Gene, protein, and drug)”. Different networks were constructed using CellDesigner 4.4.2 [24], a modeling tool that enables us to graphically represent interactions using a well-defined and consistent graphical notation in the Systems Biology Markup Language (SBML) [25,26]. The schematic diagram for the complete methodology is shown in Appendix A. The standard notation scheme for the graphical representation is provided in Appendix A.

### 2.1. Extraction of the Literature Information Related to the T. cruzi Pathways

We extracted the names of the *T. cruzi* pathways using the keywords “*T. cruzi*” AND “pathway” on Google Scholar and PubMed. A total of 70,600 hits and 1142 hits were found on Google Scholar and PubMed, respectively. We shortlisted 46 unique pathways of *T. cruzi* from this total number of hits (Appendix A). All 46 pathways were classified, based on their function (Appendix A).

Further, we extracted the abstracts related to these 46 pathways from Google Scholar and PubMed using the keywords “*T. cruzi*” AND “<Name of pathway>” through deep curation strategies. Manual curation was performed using extracted abstracts and the text describing the gene/protein or any other molecules involved in the pathway was retrieved during the curation.

### 2.2. Retrieval of the T. cruzi Molecules and Their Function

We retrieved 19,607 genes of *T. cruzi* CL Brener (*Tc*-CLB) from the NCBI database [27] (Appendix A). A total of 1756 unique genes were identified after removing the duplicate genes (Appendix A). To find out the function of the genes, an extensive literature search was carried out on PubMed and Google Scholar using the key terms “*T. cruzi*” AND “gene name”. All curated molecular evidence is summarized in Appendix A. In addition, 86,990 abstracts (Google Scholar-85,600, PubMed-1390, as of February 2021) were collected using the literature mining approach (Appendix A). The 1756 unique genes with their known molecular interactions were used to construct a comprehensive molecular map (Figure 1).

Next, a total of 19,757 *Tc*-CLB proteins were retrieved from the UniProt database [28] (Appendix A). Out of this list, 3109 unique proteins were filtered manually (Appendix A). An extensive literature search was carried out in PubMed and Google Scholar using the key terms “*T. cruzi*” AND “protein name”. All curated molecular evidence with its interactions is outlined in Appendix A. A total of 103,868 abstracts (Google Scholar-96,300, PubMed-7568), as of February 2021, were screened to obtain the molecular information related to *T. cruzi* (Appendix A). All unique proteins from the UniProt database, with their interaction information, were used for the construction of the molecular map.

### 2.3. Retrieval of the *T. cruzi* Drugs

For the construction of the *T. cruzi* drug network, we screened research articles manually, using the key terms “*T. cruzi*” AND “drug”. The 1693 abstracts with the terms “*T. cruzi*” AND “drug” were downloaded from Google Scholar and PubMed (Appendix A). The molecular evidence describing the functions of the drugs were extracted and then used for the construction of the network using CellDesigner 4.4.2 [24]

#### 2.3.1. Networks Analysis

The network topological parameters were analyzed using a Cytoscape plugin called “Network Analyzer” [29]. It calculates several parameters, such as the number of nodes, number of edges, the average number of neighbors, network diameter, network radius, characteristic path lengths, clustering coefficient, network density, network heterogeneity, network centralization, and connected components. 

#### 2.3.2. Gene Ontology (GO) Analysis

The unique *T. cruzi* genes retrieved from various sources were used for the gene ontology analysis. The gene ontology was carried out for four categories i.e., biological process, molecular function, cellular component, and metabolic pathways. The enrichment analysis of the gene ontology (GO) [30], KEGG [31], and MetaCyC pathways [32] was performed for the *T. cruzi* genes using TriTrypDB v44 [33,34]. In TriTrypDB, the data is available for a wide range of kinetoplastids. For this analysis, we selected the *T. cruzi* organism to filter out only the *T. cruzi* genes (see https://tinyurl.com/TriTrypDB). Further, the significantly enriched GO terms were clustered, summarized, and visualized using REVIGO [35]. The GO terms and pathways enrichment were considered statistically significant when the false discovery rate or FDR was less or equal to 0.05.

#### 2.3.3. Homology Modelling of the *T. cruzi* Proteins

We retrieved the FASTA sequences of 19,607 *Tc*-CLB proteins from NCBI. Since only 127 crystal structures were available in the PDB for *T. cruzi*, we decided to construct three-dimensional (3-D) structures with the help of homology modelling. As a first step, we were able to find 4905 *Tc*-CLB proteins showing a similarity (≥35%) with the sequences available in the PDB. Next, we used the MODELLER tool [36], for the homology modeling of the proteins [37].

#### 2.3.4. Molecular Docking

We developed a new algorithm named drug-dock for the large-scale automated docking using Auto dock Vina [38]. This system is based upon Python, JavaScript, and Open Babel. The system also has a module for the prediction of protein structures using homology modeling [39] in case the target structure is not present in the PDB. We obtained the structural information on the genes/proteins implicated in *Tc*-CLB from the PDB. The structure information for Bz, Nf, orlistat, and aspirin, as well as other drugs was obtained from DrugBank [40] and their side effects were retrieved from the SIDER database. The PDB format of the protein structure was converted to PDBQT format before commencing the docking procedure. For the prediction of the binding site, we have incorporated the P2Rank tool [41] in the pipeline. The system performs the generation of a configuration file for each protein, converts the structure into PDBQT format, changes the ligand from SDF to PDBQT format using Open Babel [42], and performs molecular docking using Auto dock Vina. The system also converts the ligand’s 2d-SDF to 3d-SDF before converting it into PDBQT format. The obtained config files for the proteins and ligands in PDBQT format are provided as input to Auto Dock Vina. Auto Dock Vina generates a log file and an output file. The log file contains the binding energies of proteins and drugs, whereas the output file contains the 3D structures of the 10 highest-scoring poses (orientations) of each drug attached to the protein. During the process of parsing, the final result is generated in the form of a matrix. 

## 3. Results

### 3.1. General Features of the Comprehensive Molecular Map of T. cruzi

A total of 86,990 abstracts related to *Trypanosoma* genes (using keywords “*T. cruzi* AND genes”) were collected from Google Scholar (85,600 abstracts) and PubMed (1390 abstracts) (published till February 2021) (Appendix A). Out of 19,607 genes available from NCBI (*Tc*-CLB dataset), only 1756 gene names were found to be unique in nature (Appendix A). Next, we extracted the information about the function, as well as the interaction information for each gene (Appendix A).

Similarly, we retrieved 103,868 abstracts (Google Scholar and PubMed) using the keyword “*Trypanosoma cruzi* AND Protein” (February 2021) (Appendix A). A list of the *Tc*-CLB proteins was collected from the UniProt database (19,757 proteins after removing the redundant hits [28]) (Appendix A). Further, we filtered the multi-copy proteins and hypothetical hits from the list to obtain 3109 unique protein names (Appendix A). Next, we extracted the interaction information for each of these proteins (Appendix A).

### 3.2. Features of the Comprehensive Map

The comprehensive molecular map comprises 2415 nodes and 1608 edges. The nodes include 190 genes, 1188 proteins, eight antisense RNAs, 201 complexes, 12 degraded, 48 drugs, 13 ions, 534 phenotypes, 22 RNAs, 42 unknown nodes, and 157 simple molecules (Appendix A). The edges were categorized into 1458 state transitions, five positive influences, 18 triggers, nine modulations, nine transport-related, seven translation-related, six transcription-related, one physical stimulation, and 92 others (Appendix A). The comprehensive molecular map is shown in Figure 1. It contains the molecules that are known to participate in important pathways, such as the ERK1/2 mitogen-activated kinase pathway, the glycogen synthase kinase 3 pathway, mTOR pathways, etc. We also found several housekeeping genes on the map, such as *bap1* (BRCA1 associated protein), *coq4* (coenzyme Q4), *dpy30* (Dpy-30 histone methyltransferase complex regulatory subunit), *mpc2* (mitochondrial pyruvate carrier 2), and *ndufv2* (NADH: ubiquinone oxidoreductase core subunit V2). The drugs reported as inhibitors of the *T. cruzi* growth were also included. These include Bz, tipifarnib (R115777), Nf, and leptomycin B.

### 3.3. Pathways of T. cruzi

Following the construction of the comprehensive map (see Section 3.2), we started collecting information on the pathways reported in the literature. Using text mining and deep curation, we found 46 unique pathway names in the literature (Appendix A). We extracted the molecules reported to be associated with these pathways. Further, we also extracted the molecular interaction information among the molecules involved in each pathway. A manually curated resource (database) containing papers or abstracts (with highlighted text to describe the role of the specific molecules) was assembled (Appendix A). A combined Systems Biology Markup Language (SBML) file was prepared for each pathway (Figure 2). In total, the pathways contained 1055 nodes that connect 43 genes, 838 proteins, one molecule involved in protein degradation, 37 drugs, 10 ions, 81 phenotypes, nine RNAs, and 36 simple molecules (Appendix A). These nodes are connected by 874 edges (Appendix A). We categorized these pathways into 17 major classes, including metabolic pathways, signaling pathways, degradative pathways, inflammatory pathways, etc. (Appendix A). As an example, we shall provide details of one of the constructed pathways (ubiquitin-proteasome pathway) in the next section (Figure 3).

Ubiquitination is an important process in eukaryotes and the ubiquitin-proteasome pathway enzymes are an important component of the protein degradation machinery. The process of degradation starts with the activation of ubiquitin (Ub), through the Ub-activating enzyme, E1, followed by the transfer of the activated Ub protein to Ub-conjugating enzymes (E2s) by the transacylation reaction. E2 transfers the Ub to the target protein substrates with the aid of the substrate-specific Ub ligases (E3s). The conjugation of a single Ub moiety is termed mono-ubiquitination and the subsequent conjugation of the Ub moieties leads to the formation of a polyubiquitin chain. This cascade of events leads the target substrate protein to the 26S proteasome for elimination [43]. To check the potential targets, we used molecular docking techniques to check the interactions of Bz and Nf (DB11820 and DB11989) against the molecules present in the ubiquitin-proteasome pathway. The molecules include glutathione peroxidase, ubiquitin/ribosomal protein S27a, ubiquitin-protein ligase, ubiquitin carboxyl-terminal hydrolase, 26S protease regulatory subunit, ubiquitin hydrolase, and ubiquitin-conjugating enzyme [44,45,46] (Appendix A).

### 3.4. *T. cruzi* Drugs and Network

An extensive literature search was carried out using PubMed and Google Scholar with the key terms “*T. cruzi* AND drug”. A total of 68,293 abstracts (Google Scholar 66,600 and PubMed 1693 hits published up to February 2021) were collected (Appendix A). Each report was curated to obtain the molecular information related to the drug. The relevant line(s) or paragraph(s) was highlighted and used as evidence for building the drug network (Appendix A and Figure 4). The drug network comprises 25 nodes including three proteins, 10 drugs, one simple molecule, five ions, nine phenotypes, and two unknown molecules that are connected via 26 edges (Appendix A). The edges represent interactions between each reactant or node. In this network, the interactions between reactants can be categorized into state transitions (19), and negative influences (Appendix A). In addition, we found 41 drugs for *T. cruzi* which are in different stages of development. For example, allopurinol, sulfasalazine, and thioridazine (TZD) are undergoing testing in the lab (experimental stages) [47,48,49,50] whereas posaconazole, ravuconazole, Nf, and Bz are in different phases of clinical testing [51,52,53,54] (Appendix A).

### 3.5. Application of the *T. cruzi* Drug Network

Treatment of CD is still limited to only two drugs, Bz and Nf [55,56]. Both are orally administered and can cause severe side effects, as well as long-term toxicity. Apart from these two drugs, naphthoquinone derivatives play an important role in DNA fragmentation, as well as in the release of cysteine proteases from reservosomes to the cytosol. This proteolytic process leads to parasite death [57] (Figure 2). Moreover, an antifungal—ravuconazole is a promising drug that is in clinical trials against CD, and used in combined therapy with Bz [58]. CYP51 (Sterol 14*α*-demethylase cytochrome P450) is an important enzyme with the trypanocidal activity responsible for ergosterol’s biosynthesis, that was identified in 1990. CYP51 inhibits the sterol synthesis, which is lethal to the parasite. Ravuconazole and posaconazole act through the coordination of nitrogen with heme iron into the binding cavity of CYP51 [59]. Studies on animal models found that posaconazole could be used for the treatment of acute and chronic CD [11]. The combination of Bz and itraconazole was shown to decrease the typical lesions (myocardial inflammation and fibrosis) associated with chronic CD and eliminate the parasites from the blood [12].

To check the interactions of various candidate drugs at the network level, we decided to use a drug dock algorithm. For instance, we docked Bz and Nf against the available 127 crystal structures of the *T. cruzi* proteins (Appendix A). We found that Nf is predicted to have the highest binding affinity with *T. cruzi* type B ribose 5-phosphate isomerase (TcRpiB) (−8.0 kcal/mol). During the literature curation, we found that the RpiB enzymes are present in the parasite whereas their homologs (RpiA) are absent. Further, TcRpiB turned out to be the only enzyme of the *T. cruzi* PPP (pentose phosphate pathway) which does not have a counterpart in higher eukaryotes. To check the potential impact of Bz and Nf on the PPP, we conducted a docking study against the members of the PPPs. As a comparison, we also studied the interactions of the controls (aspirin and orlistat) against the same targets. We found that the binding energy distributions are higher in the Bz and Nf study group, when compared with aspirin and orlistat. This holds, not only in the PPP of *T. cruzi*, but also in other pathways, suggesting differential binding preferences of Bz/Nf (Appendix A).

To study the potential side effects of Bz and Nf on humans, we conducted a large-scale docking study on the human proteome. For this, we collected 24,391 human proteins from the AlphaFold database [60] and performed docking with Bz. Due to technical issues, we could only dock 19,523 proteins with Bz. Interestingly, we identified biotin-protein ligase (Appendix A) as one of the top-ranking interactors of Bz (−9.1 kcal/mol of binding energy). Biotin is a water-soluble vitamin that belongs to the vitamin B complex and is an essential nutrient of all living organisms from bacteria to man [61]. In eukaryotic cells, biotin functions as a prosthetic group of enzymes, collectively known as biotin-dependent carboxylases that catalyze the key reactions in gluconeogenesis, fatty acid synthesis, and amino acid catabolism [61]. Biotin protein ligase (BPL) is required for the covalent attachment of biotin to biotin-dependent enzymes [62]. The clinical features of biotin deficiency include rashes, brittle hair, lethargy, hallucination, sleep disturbances, myalgia, and paraesthesia. The human biotin protein ligase (UniProt ID: P50747) is associated with glutamine deficiency and congenital phenotype [63] (OMIM database). Glutamine also contributes to the normal intestinal barrier function and can become deficient in some intestinal diseases, including Crohn’s disease, diarrheal illness, and short gut syndrome [64]. According to Viotti et al., the side effects of Bz vary from person to person [65]. The major side effects of Bz include insomnia, fatigue, anorexia, headache, furred tongue, gastrointestinal disturbances, skin rash, pruritus, erythema multiforme, and toxic epidermolysis [66,67]. We see a significant overlap between the clinical features of biotin deficiency and the side effects of Bz. There is a strong possibility that some of Bz’s side effects (i.e., gastrointestinal disturbance, sleep disturbances, etc.) can be linked to the off-target binding with BPL.

We further identified prostaglandin F synthase (PGF) (PDB: 4GIE) from *T. cruzi* bound to NADP, as another top-ranking target (−8.4 kcal/mol with benznidazole). The function of PGF is to catalyze the reduction of aldehydes and ketones to their corresponding alcohols. In humans, these reactions take place mostly in the lungs and the liver [68]. It is pertinent to note that PGF is involved in essential lipid-metabolism pathways in protists. Whereas in humans, PGF (Uniprot P42330) can interconvert active androgens, oestrogens, and progestins with their cognate inactive metabolites [69]. In humans, prostaglandins (PGs) E2, and PGF2α are produced in the endometrium and are important for menstruation and fertility [70]. Prostaglandin F2α synthase or old yellow enzyme (OYE), another NAD(P)H flavin oxidoreductase, similar to mitochondrial NADH-dependent type-I nitroreductase (NTR I), has been implicated in the activation pathway of other trypanocidal drugs, such as Nf but not Bz [71]. Different studies have shown that OYE was found to be downregulated in resistant parasites [72,73]. Murta et al. (2006) found that this protein was downregulated in resistant parasites, due to the deletion of three copies of the gene [73]. Likewise, by proteome analysis, it was found that OYE was under-expressed in resistant parasites [72]. Thus, PGF can be considered an important target for Bz. One of the reported side effects of Bz is hepatitis, which could be linked to the unwanted interaction of Bz with human PGF.

Next, we docked 1516 FDA-approved drugs with PGF, intending to find additional candidate drug molecules against CD (Appendix A). We found that the top three drug molecules were rifabutin, lurbinectedin, and amphotericin B. Further, we predicted 4905 structures of *T. cruzi* using homology modelling, since only 120 experimentally known structures were available in the PDB. Using this dataset, we found that the binding energy scores of Bz and Nf were significantly different from the binding energy scores of unrelated controls, aspirin, and orlistat (Welch’s T-test) (Appendix A). The top five ranking protein targets for Bz were found to be: hypothetical protein (XP_813710) (showing a similarity with the putative dynamin family), spermidine synthase (XP_816871), apurinic/apyrimidinic endonuclease (XP_816327), trans-sialidase (XP_802286), and peroxisome biogenesis factor 1 (XP_809676). Next, we used these top five targets to screen FDA-approved drugs to find potential drug candidates.

In a similar approach, the top five protein targets identified using Nf, were PIF1 helicase-like protein (XP_820585), hypothetical protein (XP_818590) (showing 99.77% similarity with RNA editing 3′ terminal uridylyl transferase 1 (*Trypanosoma cruzi*) (GenBank ID: PWV06812.1)), kinesin (XP_817032), ras-related protein Rab21 (XP_804862), and serine/threonine-protein kinase (XP_804576).

### 3.6. Therapeutic Implications of the *T. cruzi* Network

To check whether Bz/Nf produce their clinical effects, due to the preferential binding to several molecules listed in the *T. cruzi* network (N), rather than the proteomes of other pathogens, we used a dataset of 4905 *T. cruzi* protein structures (P). In addition, we created datasets of randomly selected protein structures from different pathogens (labelled as P1, P2,…Pn) as controls. We used proteins from *Plasmodium falciparum* (P1), *Mycobacterium tuberculosis* (P2), *Leishmania donovani* (P3), and *Salmonella typhi* (P4). We found that the binding energy distributions of Bz/Nf are significantly higher, when compared with the protein datasets selected from other pathogens (Welch’s T-test) (Appendix A). These results indicate that parasite-specific drugs (Bz/Nf) have specific binding affinities towards the parasite-specific proteins (i.e., *T. cruzi*). The reason could be attributed to the presence of specific types of residues and patterns/motifs in the binding sites of several parasite proteins.

Our research group has a long-standing interest in Tc24 (flagellar calcium-binding protein of 24 kDa). This protein has been proposed as a candidate for an immunotherapeutic vaccine, as well as a drug target [74]. Following the retrieval of the protein structure of Tc24 (accession ID: Q1L1I2_TRYCR) from AlphaFold [75], we conducted a large-scale docking study against Tc24 using drug molecules derived from the Zinc database [76] and a set of FDA-approved drug molecules. We found the drugs dutasteride, sirolimus, and candicidin to show the highest binding affinities against Tc24 (Appendix A).

Additionally, we conducted the molecular docking of Bz and Nf against Tc24 using Autodock Vina [38]. The PDB file for Tc24 was obtained from AlphaFold while the ligands (drugs) were obtained from DrugBank. The input PDBQT files for the protein and ligands were obtained from Autodock Vina [38]. The binding affinity for the top-scoring pose of Bz was observed as −6.9 kcal/mol while for Nf, it was observed as −6.7 kcal/mol. We also observed the distance of other modes from the best modes, in terms of the root mean square deviation (RMSD) (Appendix A).

### 3.7. Quantitative Analysis of the Networks

The topological analysis of all constructed networks was performed using a network analyzer tool [29] (Appendix A). The details were as follows:

(A) Clustering coefficient—If node A in a network is connected to node B, and B is connected to node C, A likely has a direct connection to C as well. The clustering coefficient can be used to quantify this phenomenon. It determines the average local neighborhood in a network [77], which varies frequently across the network [78]. If the clustering coefficient is near 0, the majority of nodes in the network have less than two neighbors, implying a tree-like topology [79]. The clustering coefficient value on our comprehensive map and modules is close to 0. The average clustering coefficient C(k), which indicates the metabolic network’s modularity [80], is another significant measure of the network’s structure. Using Network Analyzer [81], we discovered that our network’s average clustering coefficient was 0, indicating the presence of a tree-like structure;

(B) Network diameter is the maximum distance between two nodes. If the network is disconnected, the average of the maximum distances between the linked components is used to calculate the diameter. The diameters for the *T. cruzi* pathway network, molecular map network, and drug network were computed to be 28, 22, and 18 units, respectively;

(C) The characteristic path length is measured as an average number of edges dissociating any two nodes in the network. The pathway network has 1929 nodes with a path length of 11.73 units, the comprehensive map has 4016 nodes with a path length of 8.33 units, and the drug network has 51 nodes with a path length of 6.91 units;

(D) The average number of neighbors indicates the average connectivity of a node in the network. We observed 2.18 for our pathway network, 2.02 for the comprehensive molecular network, and 2.03 for the drug network;

(E) The network density determines compactness, which can be simply defined as the ratio of observed edges to the number of possible edges for the given network. The value ranges from 0 to 1, the closer the value, the denser and more cohesive the nodes in the network. We have computed the network density for our network and the average network density for all three networks is 0.027.

### 3.8. Gene Ontology Analysis of the *T. cruzi* Genes

To understand the biological functions of the *T. cruzi* genes used in the network construction, we performed a gene ontology analysis using TriTrypDB [33]. A total of 803 genes were recognized as belonging to the Trypanosomatidae class (Appendix A). We included only 72 genes that belong to two *T. cruzi* strains (CL Brener Esmeraldo-like (40), and Non-Esmeraldo-like (32)) (Appendix A). Based on function, the genes were enriched in a cellular component, molecular function, biological processes, and metabolic pathways. The cluster representation was performed using the REVIGO tool [35], which allows the clustering of semantically similar gene ontology terms, and labels each cluster with a single representative gene ontology term. The enriched metabolic pathway was considered statistically significant when the false discovery rate (FDR) was less than or equal to 0.05. The details of the enriched genes are provided in Appendix A.

The *Tc*-CLB Esmeraldo-like genes were enriched in several molecular functions, such as cystathionine beta-synthase activity (GO:0004122), catalytic activity (GO:0003824), and metal ion binding (GO:0046872), different cellular components, such as the intracellular membrane-bounded organelle (GO:0043231), membrane-bounded organelle (GO:0043227), and mitochondrial inner membrane (GO:0005743), and the biological processes, such as alpha-amino acid biosynthesis (GO:1901607) or cellular amino acid biosynthesis (GO:0008652) (Appendix A).

Similarly, the GO of *Tc*-CLB Non-Esmeraldo-like showed that the highest number of enriched genes were involved in different cellular components, such as the kinetochore (GO:0000776), chromosomal region (GO:0098687), and the site of double-strand breaks (GO:0035861), molecular functions, such as cystathionine beta-synthase activity (GO:0004122), catalytic activity (GO:0003824), and hydro-lyase activity (GO:0016836), and in biological processes, such as sulphur compound biosynthetic process (GO:0044272), cellular process (GO:0009987), cysteine metabolic process (GO:0006534), and cysteine biosynthetic process(GO:0019344), (Appendix A). Many pathways, such as UDP-alpha-D-galactose biosynthesis, UDP-alpha-D-glucofuranose biosynthesis, D-galactose detoxification, and puromycin biosynthesis, were enriched in *Tc*-CLB Esmeraldo-like and Non-Esmeraldo-like (Appendix A).

## 4. Discussion

This work combines the data from heterogeneous databases, including the literature, structure, and expression to construct a comprehensive map of *T. cruzi* molecules. It attempts to explain the interactions of the drug molecules in the context of networks. An effective drug molecule is expected to target the key molecules of the pathogen, as well as disrupt the key sections of its molecular network. Historically, molecular docking at a large scale has been deployed sparingly. Gao et al. used ~1100 targets [82]; Hui-Fang et al., used 1714 targets [10,83,84].

The major principle in drug discovery is to design maximally selective ligands to act on individual drug targets. However, several drugs act via the modulation of multiple proteins rather than single targets, suggesting the role of network-based therapeutics [85]. Here we performed the docking of trypanocidal drugs (Bz and Nf), control drugs (orlistat and aspirin), and 1500 FDA-approved drugs with *T. cruzi* network proteins (4905), as well as with the whole human proteome (19,523). We used our new drug repurposing pipeline to conduct large-scale docking [86]. Based on our prediction, we propose that the trypanocidal drugs show a preferential binding with *T. cruzi* proteins, as compared to the control drugs. It has been observed that benznidazole, not only binds to known targets, but also to several other targets of human proteins, which could explain some of the side effects of Bz. The overlap of the Bz side effects and biotin deficiency symptoms suggests a possibility of attenuating the side effects by the biotin administration [87,88]. Here, it is important to mention that another organic compound, “benzimidazole”, was studied by Woolley (1944). Benzimidazole derivatives have been reported to contain the trypanocidal activities [89]. Woolley (1944) reported that the similarity of the symptoms observed in animals receiving benzimidazole to those seen in biotin deficiency, suggested that the action of benzimidazole might be related to its structural similarity to biotin [90]. Both Bz and benzimidazole contain a common chemical nitrogen-containing ring.

We also observed that therapeutically unrelated drugs, i.e., orlistat and aspirin, displayed different binding patterns to the *T. cruzi* network proteins. Further studies are needed on other pathogens, which could explain the therapeutic effect of drugs at the network level. For example, similar studies could be conducted to study the binding of chloroquine against the proteome of malarial parasites. The other interesting potential application of our study is to compare the distribution of the binding energies of Bz/Nf in other related strains and species of *Trypanosoma*.

In our docking study, Nf was predicted to bind with *Tc*RpiB, with a binding affinity of −8.0 kcal/mol. *Tc*RpiB plays an important role in the pentose phosphate pathway (PPP). It is responsible for the production of nucleotide precursors and NADPH, which provide protection to trypanosomatids during oxidative stress [91]. A study performed by Loureiro et al. showed that RpiB silencing in *Trypanosoma brucei* reduced the in vitro growth of the parasites. Furthermore, RpiB silencing in the infected mice, exhibited lower parasitaemia and prolonged survival compared to control mice [92]. The absence of the RpiB enzyme in humans and its pivotal role in PPP makes it a potential chemotherapeutic target for trypanocidal drugs. RpiB is reported to be conserved among different *Trypanosoma* species [92]. Larkin et al. conducted a protein sequence alignment using ClustalW [93] and found a 67% identity for *T. brucei* RpiB versus *Tc*RpiB, and both proteins show no similarity with human ribose 5-phosphate isomerase A. Faria et al. compared the RpiB sequences of *L. infantum* (*Li*RPIB), *L. major* (*Lm*RPIB), *T. brucei* (*Tb*RPIB), and *T. cruzi* (*Tc*RPIB). They found that *Li*RPIB displays a 93% sequence identity with *Lm*RPIB and around 50% with RpiB from trypanosomes [94].

Biological systems are robust in the way that they restore the perturbations caused by drug treatments. One of the key avenues for a successful therapeutic drug or vaccine development is to overcome the biological robustness, maintained through positive or negative feedback loops of the drug/vaccine target proteins. The expression of several genes is believed to be altered during the disease phase. The genome-wide transcriptional profiling should enable us to specifically monitor the expression changes of the drug targets induced by their inhibitors or activators. The next version of this network will include the integrated genome-wide expression datasets, to study the perturbation induced by the therapeutic agent (drugs, vaccines, etc.) on *T. cruzi*.

Our research group has a long-standing interest in Tc24 (flagellar calcium-binding protein of 24 kDa). This protein has been proposed as a candidate for an immunotherapeutic vaccine, as well as a drug target. We also conducted a large-scale docking study against Tc24 using drug molecules derived from the Zinc database and a set of FDA-approved drug molecules. We found the drugs dutasteride, sirolimus, and candicidin have shown the best binding affinities against Tc24. (Appendix A).

Considering the wide variety of factors affecting the CD pathophysiology, we believe that a *T. cruzi* comprehensive map will act as a useful tool to provide information extracted from gene expression experiments, protein-protein interaction data, drug information, and clinical data information. Moreover, this will also advance our research group’s effort to use the ‘systems vaccinology’ approach to develop safe and effective vaccines against neglected tropical diseases, including CD [95]. For instance, Querec et al., 2008 used a systems biology approach to identify the early gene ‘signatures’ that predicted the immune responses in humans vaccinated with the yellow fever vaccine YF-17D [96]. Similarly, Nakaya et al. (2011) found that in subjects vaccinated with the trivalent inactivated influenza vaccine, early molecular signatures correlated with and could be used to accurately predict later antibody titers in two independent trials [97]. Li et al., (2014) performed a large-scale network integration of publicly available human blood transcriptomes and systems-scale databases in specific biological contexts, and deduced a set of transcription modules in the blood [98]. Those modules revealed distinct transcriptional signatures of antibody responses to different classes of vaccines, which provided key insights into primary viral, protein recall, and anti-polysaccharide responses [98]. These examples demonstrate the power of network-based approaches to predict immunogenicity and provide new mechanistic insights about vaccines.

## 5. Conclusions

A formalized depiction of the biological pathways is increasingly recognized as a crucial requirement for the exchange of the pathway data, modeling of their activity, and systems-level interpretation of biological data. However, there are just a handful of examples of large pathway diagrams constructed using a formalized graphical modeling language, such as SBML. The model of the *T. cruzi* pathways presented here is the most comprehensive pathway of its kind published to date. Although a time-consuming and laborious exercise, the act of converting the literature-derived knowledge into a formalized computational model is essential if we wish to truly gain a systems-level understanding of any cellular system. The *T. cruzi* pathways presented here summarize the results of years of investigations and have allowed the thorough testing of the notation system used to depict it. Furthermore, we performed large-scale molecular docking using our in-house pipeline to identify potential vaccine and drug targets. Our analysis could identify potential targets, such as type B ribose 5-phosphate isomerase, flagellar calcium-binding protein, and prostaglandin F synthase. These proteins are extensively reported in the literature as potential vaccine and drug targets [91,99,100,101]. 

## Figures and Tables

**Figure 1 vaccines-11-00267-f001:**
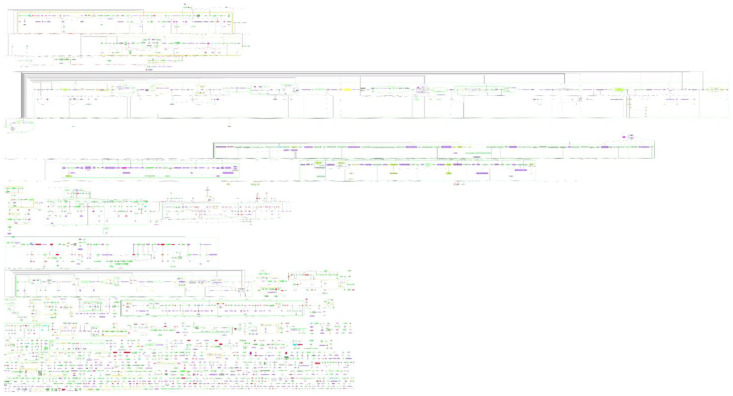
The network of the comprehensive molecular map which includes 1756 genes and 3109 proteins related to *T. cruzi*. The network is generated using CellDesigner v.4.4.2. Since the figure is very large in size, we are providing alternative links in the form of PNG on our website for the user to download and view (https://tinyurl.com/supplementary-data).

**Figure 2 vaccines-11-00267-f002:**
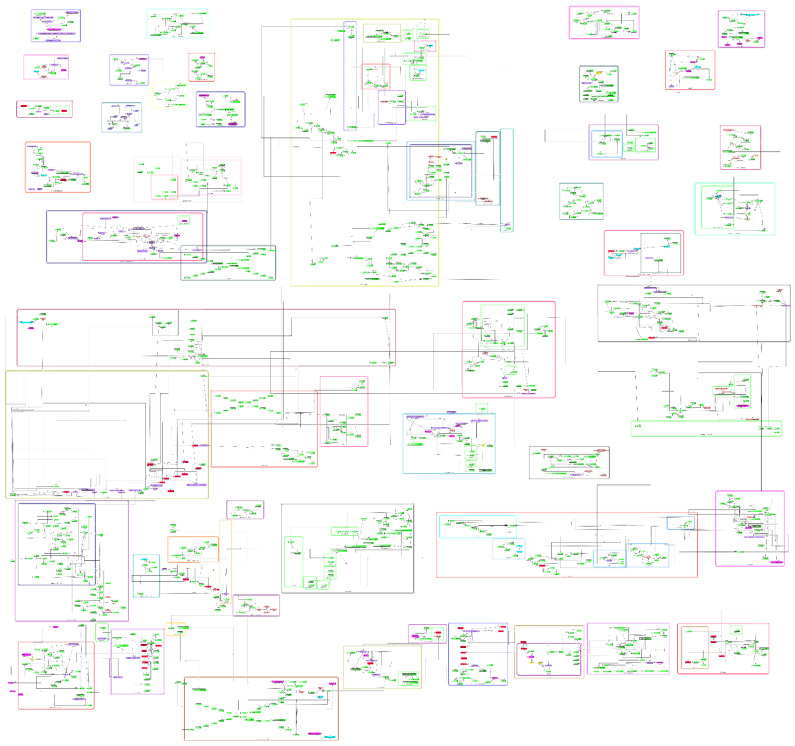
The network of molecules (gene, protein, drugs) involved in the 46 pathways. The network is generated in the CellDesigner software v.4.4.2. Since the figure is very large in size, we are providing alternative links in the form of PNG on our website for the user to download and view (https://tinyurl.com/supplementary-data).

**Figure 3 vaccines-11-00267-f003:**
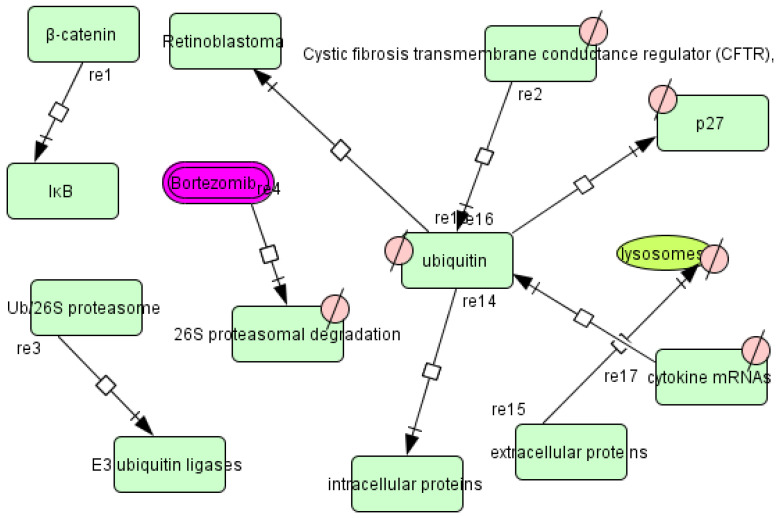
The network of the ubiquitin-proteasome pathway developed using the CellDesigner software (version—4.4.2).

**Figure 4 vaccines-11-00267-f004:**
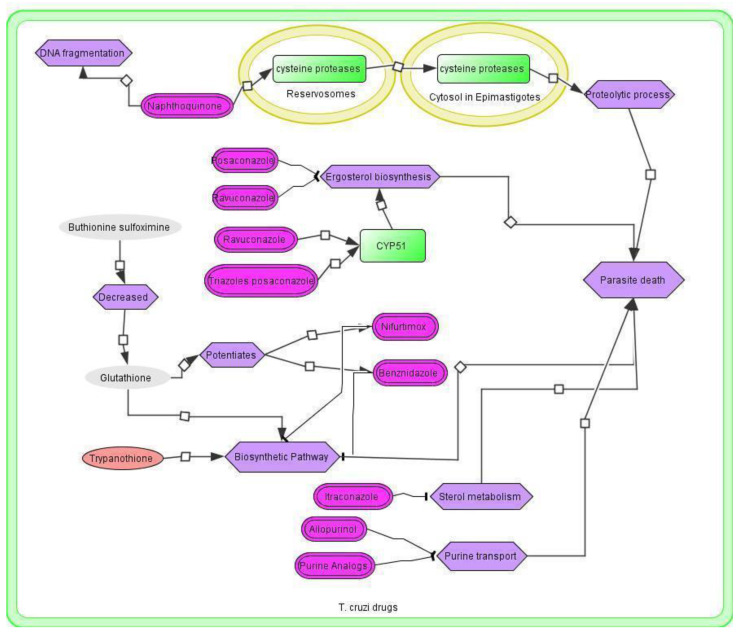
The drug network of *T. cruzi* developed using CellDesigner v.4.4.2.

## Data Availability

Data is available in the supplementary materials. In addition, data can be accessed on a dedicated website (https://tinyurl.com/Tcruzipathwaymapx).

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
