# Peer review of "A Data-Driven Approach to Construct a Molecular Map of Trypanosoma cruzi to Identify Drugs and Vaccine Targets"

_vaccines, 2023, doi:10.3390/vaccines11020267_

Round 1
Reviewer 1 Report
The article entitled "A data-driven approach to construct a molecular map of Trypanosoma cruzi to find and vaccine targets" is interesting the authors show an extensive bioinformatic analysis for searching the candidates drugs against Trypanosoma cruzi and this study is of great relevance in the field of parasitology.
In addition, the authors should show the interaction by Docking Molecular with Benznidazole and Nifurtimox and TC24 and the interaction with other proteins of T. cruzi, which is important to show the specific interaction with the proposed molecule and include it figure in the principal text of the manuscript.
Author Response
As per your suggestion, we have conducted molecular docking of Benznidazole and Nifurtimox against Tc24.
Point 1: The article entitled "A data-driven approach to construct a molecular map of Trypanosoma cruzi to find and vaccine targets" is interesting the authors show an extensive bioinformatic analysis for searching the candidates drugs against Trypanosoma cruzi and this study is of great relevance in the field of parasitology.
In addition, the authors should show the interaction by Docking Molecular with Benznidazole and Nifurtimox and TC24 and the interaction with other proteins of T. cruzi, which is important to show the specific interaction with the proposed molecule and include it figure in the principal text of the manuscript.
Response 1: We performed molecular docking of TC24 protein with Benznidazole (Bz) and Nifurtimox (Nf) drugs using Autodock Vina [Trott et al., 2010]. We found that Bz had shown binding energy as -6.9 kcal/mol whereas Nf had shown -6.7 Kcal/mol (see Supplementary Table 26 and Supplementary Figure 3).
References
1. Trott O, Olson AJ. AutoDock Vina: improving the speed and accuracy of docking with a new scoring function, efficient optimization, and multithreading. Journal of computational chemistry. 2010 Jan 30;31(2):455-61.
Reviewer 2 Report
My general perception towards this kind of studies (based on analysis of available literature) is negative, but in this particular case I see that it indeed extends and summarizes the existing knowledge. Nevertheless, the manuscript needs a serious work before it can be considered for publication in Vaccines, as the presentation of data is subpar. I noticed many annoying flows and inconsistences that must be fixed throughout the text.
Some (certainly, not all) specifics.
1) Make sure species and generic names are ALWAYS Italicized. See lns 2-3 and many more.
2) Please replace the word Protozoa and all its derivates by Protists and all its derivates.
3) For CellDesigner please cite Funahashi et al, 2003 and 2008.
4) Why Figures 5 and 6 precede Figure 1??
5) Please be more explicit about CLB and non-CLB strains. Most of the analysis is strain-specific.
6) Fig. 1 is not informative.
7) Fig. 2 belongs to the supplements.
8) Figs. 3 and 4 are not readable at all. What's the point of including them?
9) All supplementary figures (but S3) are of extremely low quality and must be replaced. By the way, why they are included in the main text?
10) Who is P.H. in the acknowledgements? P.J.H.?
11) The references MUST be reformatted: Italics for species and generic names, spelling and presentation (Italics or not) of Journal names is inconsistence, etc. Please unify.
Author Response
We have made all the necessary changes as suggested by you.
Response to Reviewer 2
Comments
Point 1: My general perception towards this kind of studies (based on analysis of available literature) is negative, but in this particular case I see that it indeed extends and summarizes the existing knowledge. Nevertheless, the manuscript needs a serious work before it can be considered for publication in Vaccines, as the presentation of data is subpar. I noticed many annoying flows and inconsistences that must be fixed throughout the text.
Some (certainly, not all) specifics.
Point 1 : Make sure species and generic names are ALWAYS Italicized. See lns 2-3 and many more.
Response 1: As per suggestion, we have revised the manuscript. The changes are highlighted in red color in the revised manuscript.
Point 2 : Please replace the word Protozoa and all its derivates by Protists and all its derivates.
Response 2: As per suggestion, we have revised the manuscript. The changes are highlighted in red color in the revised manuscript.
Point 3 : For CellDesigner please cite Funahashi et al, 2003 and 2008.
Response 3: As per suggestion, we have edited the citation.
Point 4 : Why Figures 5 and 6 precede Figure 1??
Response 4: As per suggestion, we have revised the manuscript and changed the order of Figures.
Point 5 : Please be more explicit about CLB and non-CLB strains. Most of the analysis is strain-specific.
Response 5: As per suggestion, we have revised the manuscript.
Point 6 : Fig. 1 is not informative.
Response 6: We have updated the figure and transferred it to supplementary figures (Supplementary Figure 1).
Point 7 : Fig. 2 belongs to the supplements.
Response 7: We have moved Figure 2 to Supplementary Figure 1.
Point 8 : Figs. 3 and 4 are not readable at all. What's the point of including them?
Response 8: We have created the new figures with better resolution.
Point 9 : All supplementary figures (but S3) are of extremely low quality and must be replaced. By the way, why they are included in the main text?
Response 9: We have created new figures with better resolution. The quality of all the supplementary figures have been improved in the revised manuscript. Please zoom (in/out) to see the details.
Point 10 : Who is P.H. in the acknowledgements? P.J.H.?
Response 10: We have revised P.H. to P.J.H
Reviewer 3 Report
Known in the field based on previous literatures:
1. Chagas disease (CD) is caused by the protozoan parasite- Trypanosoma cruzi, which is transmitted to animals and people by insect vectors.
2. Network pharmacology is an approach capable of describing complex relationships among biological systems, drugs, and diseases from a network perspective. Thus, molecular networks provide important avenues for the identification of drug targets, vaccine targets, and biomarkers.
In this manuscript authors reported following findings:
I have gone through the manuscript titled "A data-driven approach to construct a molecular map of Trypanosoma cruzi to find drug and vaccine targets’. Manuscript assembled an interactive network of complex interactions that occur between molecules within T. cruzi. Authors used a hybrid approach combining deep curation and text mining strategies to curate information related to T. cruzi pathways. Authors are reported following findings-
1. Authors attempted to explain the network of ubiquitin proteasome pathway and interactions of a drug molecule in the network of T. cruzi using the cell designer software.
2. Authors conducted docking studies of Benznidazole (Bz) and Nifurtimox (Nif) against the predicted protein structures using drug-dock pipeline to identify molecular targets of Bz and Nif in the T. cruzi molecular network.
The data presented are interesting and generally supportive of the conclusions drawn. There are, however, several issues including experimental validation which is not part of the article. The following minor suggestions if incorporated could help in the better understanding of the significance of the work and implications.
Minor Concerns:
1. It’s very difficult to read the network of molecular map even after zoom in figure- 3, figure- 4, and supplementary figure 1A. Authors can keep these enlarge files in supplementary.
2. As authors mentioned, there are already several existing molecular network for T. cruzi. Explain, how your study is different from rest and how does it address a specific gap in the field?
Author Response
We have made the necessary changes as suggested by you.
Response to Reviewer 3
Comments
Point 1: It’s very difficult to read the network of molecular map even after zoom in figure- 3, figure- 4, and supplementary figure 1A. Authors can keep these enlarge files in supplementary.
Response 1: We have uploaded the enlarged files of all the figures including Figure 3 and Figure 4 in PNG format and Supplementary Figure 1A in TIFF format. The clarity and resolution of the figures improve when zoomed.
NOTE: The serialization of the figures have changed. Figure 3 has been updated to Figure 1 and Figure 4 has been updated to Figure 2. Supplementary Figure 1A has been updated to Supplementary Figure 4A.
Point 2: As authors mentioned, there are already several existing molecular network for T. cruzi. Explain, how your study is different from rest and how does it address a specific gap in the field?
Response 2: The molecular map we have created is based on the fundamental mining and curation of multiple papers, new and old, for T. cruzi CL Brener strain. The reconstructed network here is designed from 46 selected unique pathways through hits obtained from PubMed and Google Scholar. The selection was based on the function of the unique gene names to their molecular interactions implicated by T. cruzi as of February 2021. The UniProt database was also incorporated to obtain unique proteins for their interactions. Such a large molecular network map has not been constructed for T. cruzi to date.
Most of the studies conducted by other authors have created networks relevant to specific biological problems. Robert et al used a constraint-based approach to assemble T. cruzi information. They utilized 51 transport reactions, 93 metabolic reactions covering energy metabolism, carbohydrate, amino acid, and four compartments (extracellular space, cytosol, mitochondrion, glycosome). Nde et al 2012 worked on the human extracellular matrix (ECM) interactome regulated by T. cruzi. Shiratsubaki IS et al. (2020) studied the Genome-Scale Metabolic Models (GEMs) of the T. cruzi and created stage-specific metabolic pathways.
In comparison, we have extracted all possible information from various types of experimental studies using manual curation as well as text mining systems. Our network is more inclusive, large, complex and heterogeneous. We have included gene expression, protein-protein interaction, drug information, and clinical information in our study. For example, our comprehensive molecular map comprises 2,415 nodes and 1,608 edges. The nodes include 190 genes, 1,188 proteins, 8 antisense RNAs, 201 complexes, 12 degraded, 48 drugs, 13 ions, 534 phenotypes, 22 RNAs, 42 unknown nodes, and 157 simple molecules (Supplementary Table 9). The edges were categorized into 1,458 state transitions, 5 positive influences, 18 triggers, 9 modulations, 9 transport-related, 7 translation-related, 6 transcription-related, 1 physical stimulation, and others (Supplementary Table 10). The network of the comprehensive molecular map is shown in Figure 2. In addition, we conducted automated large-scale molecular docking using our in-house pipeline using a large number of drug molecules and proteins.
References
1. Roberts SB, Robichaux JL, Chavali AK, Manque PA, Lee V, Lara AM, Papin JA, Buck GA. Proteomic and network analysis characterize stage-specific metabolism in Trypanosoma cruzi. BMC Syst Biol. 2009 May 16;3:52. doi: 10.1186/1752-0509-3-52. PMID: 19445715; PMCID:
PMC2701929.
2. Shiratsubaki IS, Fang X, Souza RO, Palsson BO, Silber AM, Siqueira-Neto JL. Genome-scale metabolic models highlight stage-specific differences in essential metabolic pathways in Trypanosoma cruzi. PLoS neglected tropical diseases. 2020 Oct 6;14(10):e0008728.
3. Nde PN, Lima MF, Johnson CA, Pratap S, Villalta F. Regulation and use of the extracellular matrix by Trypanosoma cruzi during early infection. Frontiers in Immunology. 2012 Nov 6;3:337.

Round 2
Reviewer 2 Report
The R1 version of the manuscript by Nath et al. showed some signs of improvement but the presentation is still very sloppy and poorly organized.
A few main comments:
1) The Figures 1 and 2 ARE NOT READABLE and (as such) are absolutely useless. Either provide readable versions or replace with smaller cuts making sense. "Zooming in" did not help.
2) Why supplementary tables are listed as non-publishable materials. This is wrong!
3) Where is Table S31?? It is mentioned in the text but it was not presented.
4) "All supplementary materials are given on our website. (https://tinyurl.com/Tcruzipathwaymap)." is not right too. They must be presented as supplements to this paper.
Minor comments.
5) Please replace ALL the instances of "protozoa" and derivatives by "protists" and derivatives. Lns 32 and 330.
6) Please Italicize species and generic names: lns 101, f.e.
7) Please revise figure legends. They are not very informative as are.
8) Please provide proper legends to the supplementary tables. Some of tehm contains unexplained graphs, etc...
